# Lupane Triterpene Derivatives Improve Antiproliferative Effect on Leukemia Cells through Apoptosis Induction

**DOI:** 10.3390/molecules27238263

**Published:** 2022-11-26

**Authors:** Lía S. Valencia-Chan, Neptis Estrada-Alfaro, Jimmy Josué Ceballos-Cruz, Luis W. Torres-Tapia, Sergio R. Peraza-Sánchez, Rosa E. Moo-Puc

**Affiliations:** 1Unidad de Investigación Médica Yucatán, Unidad Médica de Alta Especialidad, Centro Médico Ignacio García Téllez, Instituto Mexicano del Seguro Social (IMSS), Calle 41 No. 439, Col. Industrial, Mérida 97200, Yucatán, Mexico; 2Unidad de Biotecnología, Centro de Investigación Científica de Yucatán (CICY), Calle 43 No. 130, Col. Chuburná de Hidalgo, Mérida 97205, Yucatán, Mexico

**Keywords:** leukemia, lupine-type triterpene, apoptosis, molecular docking

## Abstract

Leukemia is one of the most frequent types of cancer. No effective treatment currently exists, driving a search for new compounds. Simple structural modifications were made to novel triterpenes isolated from *Phoradendron wattii*. Of the three resulting derivatives, 3α-methoxy-24-hydroxylup-20(29)-en-28-oic acid (**T1m**) caused a decrease in the median inhibitory concentration (IC_50_) on the K562 cell line. Its mode of action was apparently apoptosis, ROS generation, and loss of mitochondrial membrane potential (MMP). Molecular docking analysis showed **T1m** to produce lower binding energies than its precursor for the Bcl-2 and EGFR proteins. Small, simple, and viable modifications to triterpenes can improve their activity against leukemia cell lines. **T1m** is a potentially promising element for future research. Clarifying the targets in its mode of action will improve its applicability.

## 1. Introduction

Cancer is the second most common cause of death worldwide, accounting for one in six deaths annually [1]. In terms of incidence, mortality, and prevalence, types of leukemia are the ten most frequent cancers [2]. Acute myeloid leukemia (AML) mainly affects people under 20 years of age, and chronic myeloid leukemia (CML) is more frequent in adults over 20 years of age [3].

Allogeneic hematopoietic stem cell transplantation is the treatment of choice for leukemia but faces serious challenges in the form of a lack of compatible donors and low efficacy when the disease has reached more advanced stages [4]. Chemotherapy is also widely used, but, as with other types of cancer, it faces the ongoing problem of acquired drug resistance in cancer cells. In addition to hindering drug effectiveness, resistance is an important disadvantage in relapse. Most antineoplastics inhibit cell proliferation, but quiescent tumor cells can still activate and worsen a patient condition. Moreover, antineoplastics do not act selectively on cancer cells, causing numerous adverse effects that can complicate treatment prognosis. New, more selective, and effective drugs are clearly needed in the fight against cancer [5,6].

An estimated 40% of anticancer drugs have been inspired by natural products. Part of the search for novel anti-cancer treatments is the isolation of bioactive substances from natural sources and the generation of structural analogs or compounds inspired by natural products [7]. Plant bioprospecting is an invaluable tool in this search for new compounds. Semi-synthetic derivatization or structural modification is widely used in the search for new drugs and aims to generate structural analogs with better biological activity and fewer side effects [7].

Triterpenes are secondary metabolites and have received close attention because they are known to have anti-inflammatory [8], antioxidant [9,10], antiviral [11], antibacterial [12,13], antifungal [14] and antineoplastic activities [9], among others. Their antineoplastic activities are of particular interest in cancer research. However, their applicability is limited by factors such as low water solubility and poor bioavailability. Attempts to address these drawbacks include derivatives produced by attachment of fragments capable of enhancing both properties (e.g., heterocyclic scaffolds) [15] and implementation of delocalized lipophilic cationic compounds such as Rho123, F16, and MKT-077 [16]. Other approaches for improving triterpene solubility and bioavailability and drug delivery systems include liposomes, emulsions, nanoparticles, and cyclodextrin-drug complexes; their application increases the possibility of using natural native (or naturally occurring) triterpenes or derivatives with minor modifications [17,18].

In a previous study, the compounds 3α,24-dihydroxylup-20(29)-en-28-oic acid (**T1**) and 3α,23-*O*-isopropylidenyl-3α,23-dihydroxylup-20(29)-en-28-oic acid (**T2**), isolated from *Phoradendron wattii* [19], were evaluated in leukemia cell lines [20]. They exhibited promising results on the chronic myeloid leukemia cell line (K562). These triterpenes exert antileukemia activity by interfering with the growth of human leukemia cells. Even simple modifications of the original structure of lupane skeleton triterpenes can improve their effect on cancer cells [21,22,23,24]. The present study objective was to structurally modify compounds **T1** and **T2** with the intent of increasing their antileukemia activity.

## 2. Results

Based on the known effects of the triterpenes 3α,24-dihydroxylup-20(29)-en-28-oic acid (**T1**) and 3α,23-*O*-isopropylidenyl-3α,23-dihydroxylup-20(29)-en-28-oic acid (**T2**) on leukemia cell lines [20], the present study aim was to introduce minor structural changes on these triterpenes to generate new compounds and evaluate these derivatives to identify the way these modifications alter their activity on leukemic cells.

Three lupane-type derivatives, 3α-metoxy-24-hydroxylup-20(29)-en-28-oic acid (**T1m**), 3α-acetyl-24-hydroxylup-20(29)-en-28-oic acid (**T1Ac**) and methyl ester of 3α,23-*O*-isopropylidenyl-3α,23-dihydroxylup-20(29)-en-28-oic acid (**T2COOMe**) (Figure 1), were synthesized and evaluated. Of particular note was that the IC_50_ of compound **T1m** was much better than that of its parent compound **T1** in both cell lines (K562 and HL60) (Table 1). In contrast, the derivatives **T1Ac** and **T2COOMe** were less active; indeed, **T1Ac** had a lesser effect than its parent compound, and **T2COOMe** had no effect at all.

Bioassays for safety assessment were run for all three derivatives using a normal cell line (Vero). The **T1m** derivative exhibited minor effects against normal cells. A comparison of its IC_50_ values in the Vero and HL60 cell lines, however, indicated it had a similar effect on viability in both, meaning its selectivity index (SI) value is less than 1. This same derivative had a stronger effect in the K562 cell line compared to the Vero line, resulting in a SI of 1.69. When evaluated with peripheral blood mononuclear cells (MNC), **T1m** exhibited much stronger selectivity on the K562 cell line (SI = 2.19) than on the HL60 line (SI = 1.26).

Since **T1m** was the only derivative that decreased leukemia cell viability and it had minor effects on normal cells, its mode of action for exerting cytotoxic effects was evaluated. The first approach was to use flow cytometry to evaluate the distribution of the different cell cycle phases on the K562 cell line at three different times (18, 24, and 48 h) and two different concentrations (1/2 IC_50_ = 19.5 and IC_50_ = 39.0 μg/mL). Changes were identified in cellular distribution (Figure 2 and Appendix A). No difference between **T1m** and the negative control was observed at the 19.5 μg/mL concentration and 18 h exposure, but at 39.0 μg/mL, there was an increase in the sub-G0 population (7.29 ± 0.05%) (Figure 2A). An increase in this population (13.30 ± 1.20%) was also observed at 39.0 μg/mL and 24 h exposure (Figure 2B). When exposure time was increased to 48 h, the sub-G0 population increased to 85.85 ± 0.15% at IC_50_, indicating no cell cycle arrest. These changes are dependent on concentration and exposure time and are proportional to the concomitant increase in the sub-G0 cell population, suggesting that apoptosis is the mode of action of **T1m**. In contrast to the positive control, etoposide (14.7 μg/mL) produced an increase in the G2 population at 18 h. This is consistent with its mechanism of action, which depends on the G2 phase of the cell cycle since it is a topoisomerase II (Top2) inhibitor, stabilizing the union of Top2 and DNA, preventing union among DNA strand fragments and correcting chromosome condensation [25].

In order to corroborate the above results, double staining was performed to quantify the cell death rate via the apoptotic pathways (positive annexin V-FITC) or necrosis (only positive 7-AAD) at 18, 24, and 48 h. At 18 h, the apoptotic cell population increased by 7.79 ± 0.12% compared to the negative control (Figure 3A and Appendix A, marked in red box), while in the positive control (etoposide), it increased by 13.34 ± 0.04%, and at the 39.0 μg/mL **T1m** concentration it increased 12.50 ± 0.65%. After 24 h, no change was observed in the negative control (7.28 ± 0.39%). There was a 17.91 ± 0.06% increase in the positive control and a 24.35 ± 1.04% increase with **T1m** at IC_50_. Significant increases were observed after 48 h in the positive control (32.60 ± 0.55%) and **T1m** at 19.5 μg/mL (11.91 ± 1.04%) and 39.0 μg/mL (27.47 ± 1.23%).

Among the early biochemical changes that can occur in the cell when entering apoptosis is increased levels of reactive oxygen substances (ROS). These were evaluated using a non-fluorescent and cell-permeable probe (chloromethyl 2′,7′-dichlorodihydrofluorescein diacetate, CM-H2DCFDA), with greater fluorescence being directly proportional to the amount of intracellular ROS generated. Flow cytometry was used to measure the ROS generated by **T1m** in the K562 cell line, compared to the negative control, at the two concentrations (1/2 IC_50_, 19.5 and IC_50_, 39.0 μg/mL) and at two exposure times (2 and 4 h) (Appendix A). These exposure times were chosen because cell death by apoptosis was observed beginning at 18 h, making it necessary to decrease exposure time to observe ROS. At 2 h exposure, the 19.5 μg/mL concentration increased ROS five-fold, while the 39.0 μg/mL concentration increased it seven-fold (Figure 4A). This is comparable to the six-fold increase in the positive control (etoposide) and the seven-fold increase in the damage control (H_2_O_2_). After 4 h, ROS increased six-fold at 19.5 μg/mL **T1m** and nine-fold at 39.0 μg/mL (Figure 4B).

Another important change during apoptotic cell death is the change in the MMP associated with the intrinsic cell death pathway, which occurs after stimuli such as ROS generation that generate changes in the mitochondrial membrane. Although this change is not exclusive to apoptosis, it is an approximation that helps to elucidate its mechanism of action. Measured by means of rhodamine 123 (Rho-123), loss of MMP was found to be concentration dependent for **T1m**. When compared with the negative control, MMP decreased to approximately 45 at 19.5 μg/mL **T1m** and to 67% at 39.0 μg/mL (Figure 5 and Appendix A).

The above responses suggested the utility of molecular docking analysis using molecular targets involved in apoptosis for chronic myeloid leukemia to better understand the possible mechanism of action of **T1m**. Seven molecular targets were used. The first was BCR-ABL, a protein kinase overexpressed in most CML patients and a protein responsible for uncontrolled CML cell growth and reproduction [26]). Tyrosine-protein phosphatase non-receptor type 11 (SHP2) plays an important role in CML since it is required to initiate and maintain BCR-ABL-mediated transformation, which is vital in leukemogenesis and hematopoiesis [27]). Auro-rakinase and mortalin are overexpressed in chronic myeloid leukemia, making them targets for inhibition, especially when conventional TKI inhibitors fail [28,29]). Vascular endothelial growth factor receptor (VEGFR)) and EGFR (vascular endothelial growth factor receptor epidermal) play key roles in leukemia cell proliferation and survival [30,31]. Finally, B cell lymphoma 2 (BCL-2) is an intrinsically anti-apoptotic protein and may contribute to CML progression [32].

The binding score for BCL-2 was near that of icterogenin, an oleanan-type triterpene inhibitor of BCL-2 (Table 2, Figure 6) [33], and that for EGFR was close to that of betulinic acid, which affects the EFGR protein (Figure 7) [34]. For the anti-apoptotic protein BCL-2, **T1m** produced lower binding energies than its source triterpene (**T1**). This is related to the lower IC_50_ for the modified triterpene (Table 1). The binding energy of **T1m** was also lower than that of betulinic acid and **T1**, which again corresponds with the experimental results. These high scores suggest the involvement of these two targets in the effects observed during the experimental analysis.

## 3. Discussion

Triterpenes are metabolites and constitute the largest group of natural products, with over 30,000 known compounds to date [35]. Because of their various biological activities (anti-inflammatory, antioxidant, antiviral, antibacterial, antifungal, and anticancer), they have received ample attention in medical research [36,37]. Of particular interest have been lupane skeleton triterpenes; for instance, betulinic acid is ubiquitous in most plants [38] and exhibits effects on melanoma cell lines, especially humans [39].

Special attention has been given to the study of structural modifications in betulinic acid, mainly at promising sites such as C-28 and C-3. For example, slight modifications such as from betulinic acid to the methyl ester of betulinic acid and from betulanic acid to the methyl ester of betulonic acid have resulted in increased activity in cell lines such as K562, SK-MEL-28, GOTO, NB-1 and B162F2 [21,23,24]. Moreover, esterifications at C-3 of lupane skeleton triterpenes, such as modification of betulinic acid, are reported to make it a promising antineoplastic agent [40]. Small modifications such as these can improve the effects of lupane skeleton triterpenes on cancer cells [41,42].

However, triterpenes, and especially native ones, have major drawbacks, principally their low bioavailability caused by low hydrosolubility. Structural modifications can be made to mitigate this drawback, such as the introduction of heterocyclic pyridine derivatives [15] or the introduction of decolalized lipophilic cationic compounds such as triphenylphosphonium (TPP), F16, and Rho123 [16]. The latter is of particular use since, in addition to improving water solubility and, consequently, bioavailability, it selectively targets mitochondria [41]. This occurs because triterpenes, such as betulinic acid and betulin, can modulate the surface properties of mitochondrial membranes since they directly affect the lipid bilayer of mitochondria membranes [43].

This is important because the intrinsic apoptotic pathway involves a variety of non-receptor-mediated stimuli that produce intracellular signals, which act directly on targets within the cell and are mitochondria-initiated events. Stimuli such as ROS generation cause changes in the inner mitochondrial membrane, opening the mitochondrial permeability transition pore. This generates lipoperoxidation, causing alterations in the membrane structure, affecting its fluidity, and damaging its integrity. This results in the loss of MMP and release of pro-apoptotic proteins into the cytosol, cytochrome c, and Smac/DIABLO, with consequent activation of cysteine-aspartic acid protease 3 (CASP3) [44,45,46,47].

Triterpenes such as lupane-type triterpenes can modulate a cascade of events through processes mediated by ROS [48]. They have membranotropic activity, increasing mitochondrial membrane permeability and promoting cell death by apoptosis [49,50,51]. In the present results, **T1m** was observed to generate ROS and thus promote the loss of MMP. This is consistent with reports that the main mode of action for native triterpenes is ROS generation and consequent damage of the mitochondrial membrane, which directly affects the functioning of the mitochondrial respiratory chain complex, leading to ROS over-generation [49].

Control and regulation of these apoptotic mitochondrial events occur through members of the BCL-2 family of proteins. Members of this family (e.g., the anti-apoptotic protein BCL-2) govern mitochondrial membrane permeability. The main mechanism of action of the BCL-2 family of proteins is believed to be the regulation of cytochrome c release from mitochondria through the alteration of mitochondrial membrane permeability [45,52]. Since BCL-2 proteins potentially block apoptosis via the intrinsic pathway and are found at elevated levels in cancers, especially blood cancers, they are promising targets for therapeutic interventions [45,46]. This is why a molecular docking analysis was performed on the BCL-2 protein using **T1m**, which has binding energy similar to icterogenin [33] and has been shown to affect BCL-2 proteins with better binding energy than its **T1** parent compound.

Overexpression of the epidermal growth factor receptor (EGFR) is involved in cancers such as leukemia and has a downstream signaling pathway associated with the BCL-2 family [53]. In leukemia, EGFR function is deregulated, playing an important role in cell survival and proliferation. As a result, EGFR tyrosine kinase activity is a promising therapeutic target when treating leukemia [54,55]. For this reason, molecular docking was used here to evaluate the binding of **T1m** in the EGFR kinase domain. The **T1m** binding energy was better than that of betulinic acid, which is known to inhibit this protein, and better than its **T1** parent compound; **T1m** is clearly a promising candidate for future research.

Apoptotic cells exhibit several biochemical modifications. Among them is the expression of cell surface markers that cause early phagocytic recognition of apoptotic cells by adjacent cells, allowing rapid phagocytosis without seriously compromising the surrounding tissue. This is accomplished by the movement of normal inward-facing phosphatidylserine from a cell’s lipid bilayer toward expression in the outer layers of the plasma membrane [56,57,58]. One way of identifying apoptosis is through the binding of annexin V, a protein that specifically interacts with and binds strongly to phosphatidylserine residues [59]. Flow cytometry was used here to quantify the percentage of cells in apoptosis.

The present results suggest **T1m** as a candidate for future research into its potential role in cancer treatments. It can lead myeloid leukemia cells to apoptosis, generating reactive oxygen substances and permeating mitochondrial membranes, as well as being a possible EGFR and BCL-2 inhibitor. Of note is that, despite its low water solubility (Log P 5.82), the SwissADME platform results (Appendix A) showed **T1m** to have a greater probability of passive absorption in the gastrointestinal tract (e.g., it is in the “white” zone) than **T1Ac** and **T2COOMe**. Compound **T1m** is also a predicted non-substrate of P-glycoprotein (PGP-), which is a positive characteristic since this drug efflux pump has been linked to the development of anticancer drug resistance via low drug accumulation in multidrug-resistant cells [60]. Drug delivery systems may be useful in improving the applicability of bioactive derivatives such as **T1m**, but further modifications are still needed to improve their physicochemical properties; for example, attachment of solubility enhancer fragments is a possible approach. However, a deep understanding of **T1m’s** mode of action is important to prevent the introduction of modifications that can block key features and hinder vital interactions. Further research can concentrate on the expression of genes and proteins (mainly Bcl-2 and EFGR), evaluation of the effects of **T1m** on leukemia stem cells, determination of its in vivo bioavailability, and finally, toxicity studies.

## 4. Materials and Methods

### 4.1. General Experimental Produces

NMR spectra were recorded in CDCl_3_ on a Bruker Avance 400 spectrometer (Billerica, MA, USA). The chemical shifts are given in δ (ppm) with residual deuterated solvent as an internal reference and coupling constants in Hz.

Precoated TLC silica gel 60 F_254_ aluminum sheets from Sigma-Aldrich were used for thin-layer chromatography (0.25 and 0.5 mm layer thickness for analytical and preparative TLC, respectively) and visualized under short (254 nm) and long (366 nm) wavelength UV light or a spray reagent (H_2_SO_4_− AcOH−H_2_O, 1:20:4). Column chromatography (CC) was conducted using silica gel 60 (63–200, 40–63, or 2–25 μm particle size) or Sephadex LH-20 from Sigma-Aldrich (Saint Louis, MO, USA).

The bioassays were performed inside a laminar flow hood, brand NuAire (Playmouth, MN, USA), class II, type A2. The cells were kept inside a CO_2_ incubator with a water jacket and HEPA filter, brand NuAire.

### 4.2. Plant Material

Aerial parts of *P. watti* were collected in March 2019 on the Hunucma-Sisal highway (21°05′38.0″ N, 89°58′21.4″ W), Yucatan (Mexico), and the plant was identified by Tech. Paulino Sima-Polanco (CICY). A voucher sample (PS 3220) was deposited at the *U Najil Tikin Xiu* herbarium of CICY. The aerial parts were dried under artificial light (50–60 °C) for 3 days and then ground. Finally, after having the plant material dry, it was crushed using a blade mill without cooling (Pagani, model 2030) until a particle size corresponding to the number 7 obtained 33.5 kg.

### 4.3. Extraction and Isolation

The compounds 3α,24-dihydroxylup-20(29)-en-28-oic acid (**T1**) and 3α,23-*O*-isopropylidenyl-3α,23-dihydroxylup-20(29)-en-28-oic acid (**T2**) were isolated from the aerial parts of *Phoradendron wattii* according to a previous study [19].

### 4.4. Preparation of Compound T1m from 1

An amount of 24.0 mg (0.04 mmol) of compound **T1** and 110 mg (0.79 mmol) of K_2_CO_3_ were solubilized in 0.5 mL of CH_3_I and 1 mL of Me_2_CO. The reaction was carried out under constant stirring for 72 h at room temperature. Once the reaction product was obtained, the product was extracted by liquid-liquid extraction with H_2_O:EtOAc (1:2 ×2 and 1:1), and the organic phase was dried with Na_2_SO_4_ and distilled to remove the solvent. The residue was purified by CC (Hexane-Me_2_CO 95:5) to give 11.9 mg (49.6%) of 3α-metoxy-24-hydroxylup-20(29)-en-28-oic acid (**T1m**).

3α-metoxy-24-hydroxylup-20(29)-en-28-oic acid (**T1m**): white, amorphous powder (CHCl_3_), R_f_ = 0.36 (Hexane-Me_2_CO 8:2); ^1^H-NMR (CDCl_3_, 400 MHz) δ 4.73 (d, *J* = 2.3 Hz, 1H), 4.59 (dd, *J* = 2.5, 1.4 Hz, 1H), 3.66 (s, 4H), 3.52 (dd, *J* = 10.5, 3.6 Hz, 1H), 3.38 (d, *J* = 11.3 Hz, 1H), 3.08–3.92 (m, 2H), 2.28–2.18 (m, 1H), 2.10 (br s, 1H), 1.98–1.81 (m, 3H), 1.68 (s, 3H), 1.66–1.54 (m, 4H), 1.54–1.30 (m, 12H), 1.29–1.12 (m, 3H), 1.04 (dd, *J* = 13.0, 4.5 Hz, 1H), 0.99 (s, 3H), 0.92 (s, 3H), 0.85 (s, 3H), 0.67 (s, 3H). ^13^C-NMR (CDCl_3_, 100 MHz) δ 176.9, 150.8, 109.7, 71.6, 56.7, 51.4, 50.5, 49.6, 47.2, 43.1, 42.6, 41.0, 40.6, 38.4, 37.2, 37.1, 34.0, 33.1, 32.3, 31.1, 30.8, 29.8, 26.6, 25.7, 21.0, 19.5, 18.2, 17.9, 16.3, 16.1, 15.0, see Appendix A.

### 4.5. Preparation of Compound T1Ac from T1

An amount of 20.0 mg (0.04 mmol) of compound **1** was solubilized in 1.0 mL of Ac_2_O and 0.5 mL of C_5_H_5_N. The reaction was carried out under constant stirring for 72 h at room temperature. Once the reaction product was obtained, the product was stopped with H_2_O and extracted by liquid-liquid extraction with H_2_O:EtOAc (1:2 ×2 and 1:1), and the organic phase was washed with HCl (5%) and brine. The organic fraction was dried over Na_2_SO_4_, filtered, and after removing the solvent under reduced pressure, the crude reaction product was purified by CC, eluting with a mixture of Hexane- Me_2_CO 95:5 to give 12.3 mg (61.5%) of 3α-acetyl-24-hydroxylup-20(29)-en-28-oic acid (**T1Ac**).

3α-acetyl-24-hydroxylup-20(29)-en-28-oic acid (**T1Ac**): white, amorphous powder (CHCl_3_), R_f_ = 0.46 (Hexane-Me_2_CO 8:2); ^1^H-NMR (CDCl_3_, 400 MHz) δ 4.82 (t, *J* = 2.5 Hz, 1H), 4.74 (d, *J* = 2.3 Hz, 1H), 4.60 (s, 1H), 4.02 (d, *J* = 10.4 Hz, 1H), 3.80 (d, *J* = 10.4 Hz, 1H), 3.00 (td, *J* = 10.7, 4.7 Hz, 1H), 2.28 (dt, *J* = 12.7, 3.0 Hz, 1H), 2.19 (td, *J* = 12.3, 3.6 Hz, 1H), 2.02 (s, 3H), 1.97 (d, *J* = 7.4 Hz, 1H). 1.88–1.77 (m, 1H), 1.69 (s, 3H), 1.67–1.60 (m, 2H), 1.58–1.15 (m, 18H), 1.14–1.04 (m, 2H), 1.02 (s, 3H), 0.97 (s, 3H), 0.94 (s, 3H), 0.88 (s, 3H). ^13^C-NMR (CDCl_3_, 100 MHz) δ 171.5, 170.5, 150.5, 109.9, 73.2, 70.7, 56.5, 50.5, 49.4, 47.7, 47.0, 42.7, 40.8, 39.7, 38.5, 37.2, 37.2, 33.9, 33.6, 32.3, 30.7, 29.8, 25.6, 22.3, 21.4, 21.0, 20.8, 19.5, 18.7, 17.3, 16.2, 15.0, see Appendix A.

### 4.6. Preparation of Compound T2m from T2

An amount of 15.3 mg (0.03 mmol) of compound **2** and 110 mg (0.79 mmol) of K_2_CO_3_ were solubilized in 0.5 mL of CH_3_I and 1 mL of Me_2_CO. The reaction was carried out under constant stirring for 72 h at room temperature. Once the reaction product was obtained, the product was extracted by liquid-liquid extraction with H_2_O:EtOAc (1:2 ×2 and 1:1), and the organic phase was dried with Na_2_SO_4_ and distilled to remove the solvent. The residue was purified by CC (Hexane-Me_2_CO 95:5) to give 11.5 mg (75%) of metyl ester of 3α,23-*O*-isopropylidenyl-3α,23-dihydroxylup-20(29)-en-28-oic acid (**T2m**).

Metyl ester of 3α,23-*O*-isopropylidenyl-3α,23-dihydroxylup-20(29)-en-28-oic acid (**T2m**): white, amorphous powder (CHCl_3_), R_f_ = 0.56 (Hexane-Me_2_CO 9.8:0.2); ^1^H-NMR (CDCl_3_, 400 MHz) δ 4.73 (d, *J* = 2.4 Hz, 1H), 4.59 (dd, *J* = 2.5, 1.4 Hz, 1H), 3.66 (s, 3H), 3.65 (d, *J* = 12.1 Hz, 1H), 3.60 (t, *J* = 3.0 Hz, 1H), 3.23 (d, *J* = 12.1 Hz, 1H), 2.99 (td, *J* = 10.8, 4.6 Hz, 1H), 2.29–2.12 (m, 2H), 1.95–1.70 (m, 4H), 1.68 (br s, 3H), 1.65–1.11 (m, 17H), 1.40 (s, 6H), 1.05 (td, *J* = 13.0, 4.5 Hz, 1H), 0.99 (s, 3H), 0.91 (s, 3H), 0.84 (s, 3H), 0.68 (s, 3H). ^13^C-NMR (CDCl_3_, 100 MHz) δ 176.9, 150.8, 109.7, 98.2, 73.2, 68.6, 56.7, 51.4, 50.5, 49.6, 47.1, 43.2, 42.6, 41.1, 38.4, 37.1, 35.3, 34.3, 33.4, 32.4, 30.8, 29.8, 29.4, 25.7, 23.8, 20.8, 19.5, 17.9, 17.5, 16.7, 16.2, 15.0, see Appendix A.

### 4.7. Cell Lines

Chronic myeloid leukemia (K562; ATCC CCL-243), acute myeloid leukemia (HL60; ATCC CCL-240), and green monkey kidney cells (Vero, ATCCCCL-81) from the American Type Culture Collection (ATCC) were provided by María Antonieta Chávez Gónzalez from Unidad de Investigación Oncológica, Hospital de Oncología, Centro Médico Nacional, Instituto Mexicano del Seguro Social (IMSS, Ciudad de México, México). The K562 cell line was maintained with RPMI medium at 10% FBS, the HL60 cell line was maintained with IMDM medium at 20% FBS, and the VERO cell line was maintained with DMEM medium at 10% FBS; 1% penicillin-streptomycin was added to both cell lines and incubated at an atmosphere of culture with 95% humidity and 5% CO_2_ at 37 °C. All evaluation was performed between 4 and 5 cell passages. 

### 4.8. Compounds and Controls

The compounds (**T1**, **T1m**, **T1Ac**, **T2,** and **T2COOMe**) were dissolved in DMSO at a concentration of 10 mg/mL.

In all cases, medium with DMSO (0.1%) was used as a negative control, and etoposide was used as positive control for K562, HL60, Vero cell lines, and normal MNC. All tests were carried out in triplicate.

### 4.9. Bioassay of Viability in Leucemic Cell Lines

A total of 2 × 10^5^ cells/well were cultured into 24-well plates and treated with different concentrations of four compounds (0.1, 1, 10, and 100 μg/mL) for 48 h. At the end of this time, the cells were collected and stained with 4′,6-diamidino-2-phenylindole dihydrochloride (DAPI) at 500 ng/mL for 30 min, according to manufacturer instructions, for 15 min in darkness. The samples were then analyzed by flow cytometry using a CytoFlex flow cytometer (Beckman Coulter, Indianapolis, IN, USA).

### 4.10. Bioassay of Viability in Vero Cell Line

The cells (1 × 10^4^ cells/well) were seeded onto 96-well plates and incubated at 37 °C and 5% CO_2_ until reaching 70–80% confluence. The medium was replaced with fresh medium containing the five different concentrations (0.1, 1, 10, and 100 μg/mL) of compounds for 48 h. After 100 μL of 10% trichloroacetic acid were added to each well, and cells were incubated for 30 min at 4 °C. Then the supernatant was removed, 100 μL of sulforhodamine B (SRB) at 0.1% with 1% of acetic acid were added, and incubated again for 15 min. Subsequently, washings were carried out with 1% acetic acid; and finally, the dye was solubilized with 200 uL of tris-base at 10 mM; and the optical density was measured at 560 nm using a GloMax spectrophotometer (Promega, WI, USA).

### 4.11. Viability Test in Normal Mononuclear Cells

Normal mononuclear cells were obtained from normal human peripheral blood monocytes and donated by the corresponding author (Moo-Puc) of the manuscript submitted. The MNC was purified with FicollPaque Plus (Pharmacia Biotech, Uppsala, Sweden) by centrifugation at 400× *g* at room temperature for 30 min according to the manufacturer’s protocol. Once the MNC were obtained, they were resuspended in RPMI medium (ATCC) with 10% FBS, and they were counted using a hemocytometer, previously stained with a trypan blue solution, verifying the viability of 95% [61].

Cells were plated at 2 × 10^5^ cells/well in 24-well plates and incubated with different concentrations of compounds **T1m** (0.1, 1, 10, and 100 μg/mL) for 48 h. RPMI with 10% FBS was used as a culture medium for cell growth. After this time, cells were collected, washed with PBS, and stained with DAPI at 500 ng/mL for 30 min, according to manufacturer instructions. Cells analysis was carried out in a CytoFlex flow cytometer (Beckman Coulter, Indianapolis, IN, USA).

### 4.12. Cell Cycle Assay

One million cells of K562 were cultured in 12-well plates and incubated for 18, 24, and 48 h with compound **T1m** at two different concentrations (19.5 and 39.0 μg/mL). After this time, the cells were collected and centrifuged at 300× *g* at room temperature for 5 min. The cells were resuspended in PBS, added absolute ethanol on ice, and incubated for 1 h at 4 °C. After that. Cells were centrifuged at 300× *g* at 4 °C for 5 min. Next, cells were washed with PBS 3% FBS and permeabilized for 20 min with 0.1% triton. After, the cells were stained with DAPI at 500 ng/mL for 1 h. After this procedure, cells were analyzed using at 500 ng/mL. Cells analysis was carried out in an Attune flow cytometer (Thermo Fisher Scientific, Carlsbad, CA, USA).

### 4.13. Apoptosis Assay

Cells (2 × 10^5^) were cultured in 24-well plates and treated with compound **T1m** at two different concentrations (19.5 and 39.0 µg/mL), to later be incubated for 18, 24, and 48 h. After this time, cells were washed with PBS and stained with annexin V-FITC and 7-AAD apoptosis kit (BD Bioscience) and incubated in the dark for 15 min. Then they were analyzed by CytoFlex flow cytometry (Beckman Coulter, Indianapolis, IN, USA).

### 4.14. Assessment of Intracellular Reactive Oxygen Species (ROS)

Cells (1× 10^5^) were cultured in 12-well plates and were stained with 10 μM of chloromethyl 2′,7′-dichlorodihydrofluorescein diacetate (CM-H2DCFDA) for 30 min at 37 °C. Briefly, the cells were treated with compound **1m** at two different concentrations (19.5 and 39.0 µg/mL) and control (H_2_O_2_ at 50 µM) to later be incubated for 2 and 4 h. After this time, cells were analyzed using CytoFlex flow cytometry (Beckman Coulter, Indianapolis, IN, USA) [62].

### 4.15. Assessment of Mitochondrial Membrane Potential (ΔψM)

Cells (1× 10^5^) were cultured in 12-well plates and treated with compound **1m** at two different concentrations (19.5 and 39.0 µg/mL), to later be incubated for 18 h. After this time, cells were stained with 5 μM of rhodamine 123 (Rho-123) and incubated for 30 min at 37 °C. Briefly, cells were washed with PBS and analyzed using a Cytoflex flow cytometer (Beckman Coulter, Indianapolis, IN, USA) [63].

### 4.16. Molecular Docking

Protein target structures for molecular docking studies were retrieved from RCSB Protein Data Bank, selecting Bcl-2 in complex with a small molecule inhibitor targeting Bcl-2 BH3 domain (*N*-(6-{4-[(4’-chlorobiphenyl-2-yl)methyl]piperazin-1-yl}-1,1-dioxido-1,2-benzothiazol-3-yl)-4-{[(2*R*)-4-(dimethylamino)-1-(phenylsulfanyl)butan-2-yl]amino}-3-nitrobenzenesulfonamide) (PDB ID: 4IEH) and Epidermal Growth Factor Receptor tyrosine kinase domain with erlotinib as an inhibitor (PDB ID: 1M17) [50]. Three-dimensional structures of tested ligands were generated in their low-energy conformation using ChemDraw Professional v.16, based on crystallographic data of the compound or a closely related structure when available, and further energy minimization with an MM2 force field. Targets were prepared by removing co-crystalized inhibitors and water molecules, then processed by the LePro tool. Molecular docking studies were performed using LeDock software [64]. A receptor grid was set for each target, restraining it from including the position of co-crystalized inhibitor and surrounding cavities (resulting search volumes shown in Table 3). Generated poses were set to 200 and 1.0 of RMSD. Selected poses and interactions were analyzed in BIOVIA Discovery Studio Visualizer v19.1.

### 4.17. Analysis of Results

All cytometric data were analyzed using FlowJo^TM^ v10.6 software. Results of viability are expressed as the concentration of agent that reduces cell growth by 50% (IC_50_), calculated by GraphPad Prism 5 software. The data were expressed as means ± SEM. Statistical significance was calculated with a one-way analysis of variance (ANOVA) followed by *Dunnett’s post-hoc* test, applying a *p* < 0.05 significance level.

## 5. Conclusions

Triterpenes isolated from *Phoradendron wattii* were slightly modified and characterized by ^1^H-NMR. Three triterpenic derivatives were generated, of which the compound 3α-methoxy-24-hydroxylup-20(29)-en-28-oic acid (**T1m**) appeared the most promising for further research. Compared to the other two compounds, it exhibited better inhibition of two leukemia cell lines (K562 and HL60) via death through apoptosis, ROS generation, and disruption of mitochondrial membrane potential. Molecular docking analysis showed **T1m** to have a higher affinity against BCL-2 and EGFR than its precursor. Overall, a simple and viable modification generated a derivative with improved biological activity against leukemia cell lines, highlighting the promise of **T1m** in future research.

## Figures and Tables

**Figure 1 molecules-27-08263-f001:**
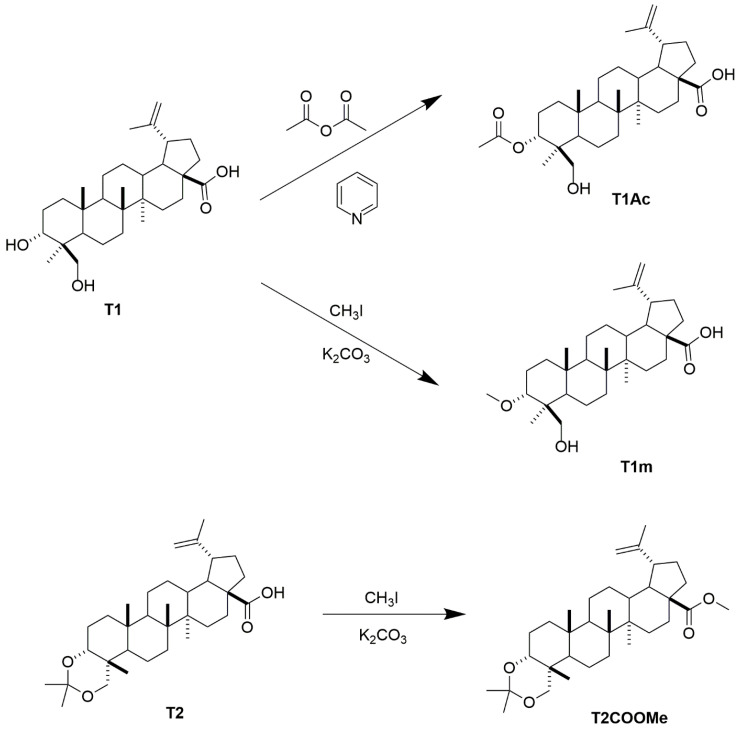
Synthesis of compounds **T1Ac**, **T1m,** and **T2COOMe**.

**Figure 2 molecules-27-08263-f002:**
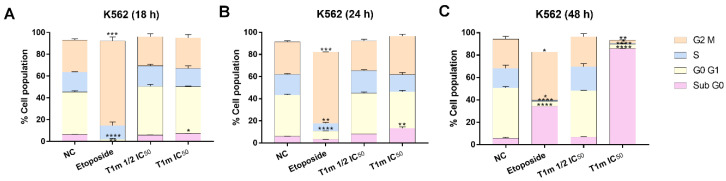
**T1m** induces increases in the sub-G0 cell population without causing cell cycle arrest. Effects of **T1m** concentration on cell cycle status in K562 cells at three different **T1m** exposure times: 18 h (**A**), 24 h (**B**), and 48 h (**C**). Data are expressed as the percentage of cell populations and represent mean ± SEM of three independent experiments of cell cycle phase distribution. A significant difference between cell cycle phases was identified using a one-way analysis of variance (ANOVA) followed by a Dunnett’s post-hoc test. Significance was set at * *p* < 0.05, ** *p* < 0.01, *** *p* < 0.001, and **** *p* < 0.0001 vs. negative control (NC).

**Figure 3 molecules-27-08263-f003:**
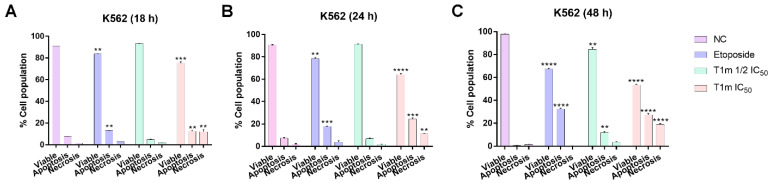
**T1m** induced apoptosis in K562 cells at both evaluated concentrations at three exposure times: 18 h (**A**), 24 h (**B**), and 48 h (**C**). The results represent the mean ± SEM of at least three independent experiments. Statistical significance was identified with a one-way analysis of variance (ANOVA) followed by a Dunnett’s post-hoc test. Significance was set at ** *p* < 0.01, *** *p* < 0.001, and **** *p* < 0.0001 vs. negative control (NC).

**Figure 4 molecules-27-08263-f004:**
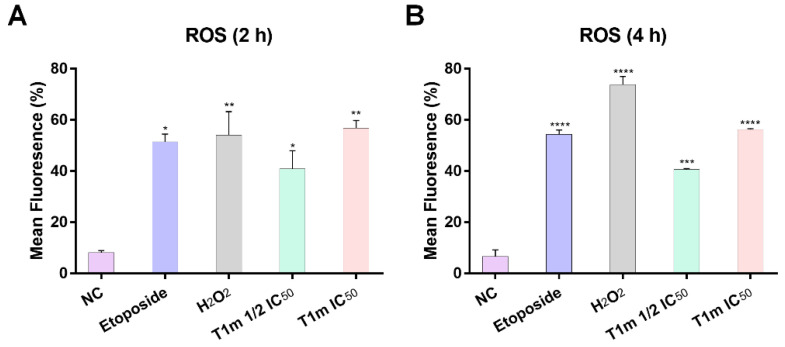
**T1m** increases ROS levels in the K562 cell line. **T1m** was evaluated at two concentrations and exposed for 2 h (**A**) and 4 h (**B**). The results represent the mean ± SEM of at least three independent experiments. Statistical significance was identified with a one-way analysis of variance (ANOVA) followed by a Dunnett’s post-hoc test. Significance was set at * *p* < 0.05, ** *p* < 0.01, *** *p* < 0.001, and **** *p* < 0.0001 vs. negative control (NC).

**Figure 5 molecules-27-08263-f005:**
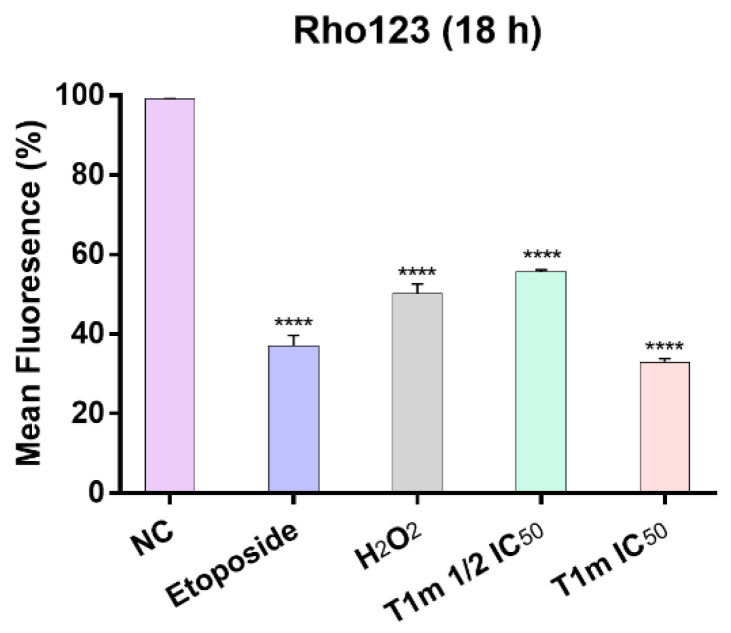
**T1m** induced loss of mitochondrial membrane potential in the K562 cell line at two concentrations. Results represent the mean ± SEM of at least three independent experiments. Statistical significance was identified with a one-way analysis of variance (ANOVA) followed by a Dunnett’s post-hoc test. Significance was set at **** *p* < 0.0001 vs. negative control (NC).

**Figure 6 molecules-27-08263-f006:**
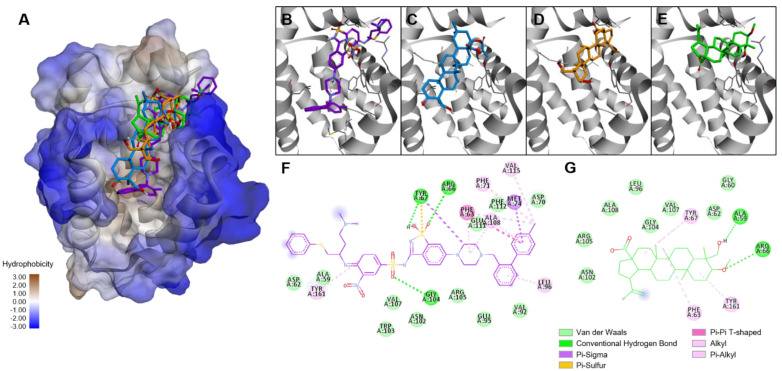
Binding modes of **T1m** in the BCL−2 protein. (**A**) Superimposed poses inside inhibitor binding site. Close−up of the binding site with (**B**) co-crystallized inhibitor (Re−docking RMSD: 3.603 Å), (**C**) icterogenin, (**D**) compound **T1,** and (**E**) compound **T1m**. Interactions of BCL−2 protein with (**F**) co-crystallized inhibitor and (**G**) **T1m**.

**Figure 7 molecules-27-08263-f007:**
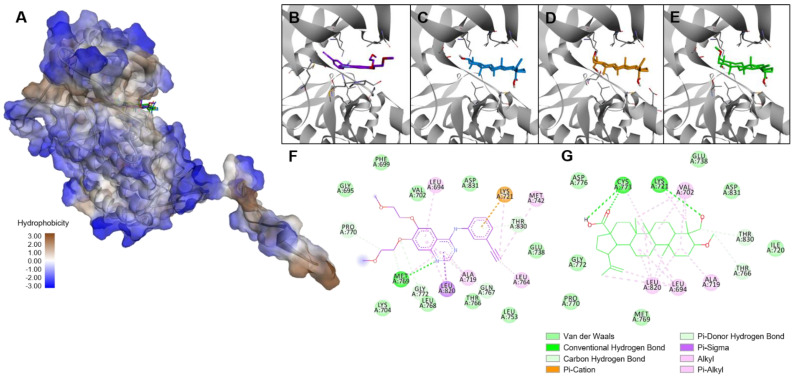
Binding modes of **T1m** in the EGFR tyrosine kinase domain. (**A**) Superimposed poses inside inhibitor binding site. Close-up of the binding site with (**B**) erlotinib (Re−docking RMSD: 1.722 Å), (**C**) betulinic acid, (**D**) compound **T1,** and (**E**) compound **T1m**. Interactions of EGFR tyrosine kinase domain with (**F**) erlotinib and (**G**) **T1m**.

**Table 1 molecules-27-08263-t001:** Mean inhibitory concentrations of the lupane skeleton triterpenes were evaluated against the K562, HL60, VERO, and MNC cell lines.

Compound	IC_50_ (μg/mL)			
	K562	HL60	VERO	MNC
**T1**	100 ± 1.0 (IS 1.37)	>100	137.2 ± 1.1	—
**T1Ac**	>100	>100	>200	—
**T1m**	39.49 ± 1.1 (IS 1.69, 2.19)	68.56 ± 1.2 (IS 0.97, 1.26)	66.76 ± 1.0	86.49 ±1.2
**T2**	>100	>100	>200	—
**T2COOMe**	>100	>100	>200	—
**Etoposide**	14.7 ± 1.2 (IS 2.00, 6.80)	1.30 ± 0.3 (IS 22.63, 76.92)	29.42 ± 1.1	>100

MNC: peripheral blood mononuclear cells; SI: selectivity index (ratio of IC_50_ in the Vero or MNC cell line over IC_50_ in the cancer cell line), —: Not tested.

**Table 2 molecules-27-08263-t002:** Docking scores for compounds **T1** and **T1m**.

Compound	Score (kcal/mol)	
	BCL-2	EGFR (TK Domain)
**T1**	−4.59	−4.78
**T1m**	−5.27	−4.91
**Icterogenin**	−5.59	—
**Betulinic acid**	—	−4.75
co-crystalized inhibitor	−10.13	−7.19

**Table 3 molecules-27-08263-t003:** Search volume for molecular docking.

PDB Entry	Search Volume					
	Minimum (Å)			Maximun (Å)		
	x	y	z	x	y	z
**4IEH**	−0.30965	18.84145	0.74902	22.77645	32.09515	19.18412
**1M17**	12.28565	−3.87359	42.20015	31.74615	11.64701	57.52825

## Data Availability

Not applicable.

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
