# Peer review of "Lupane Triterpene Derivatives Improve Antiproliferative Effect on Leukemia Cells through Apoptosis Induction"

_molecules, 2022, doi:10.3390/molecules27238263_

Round 1

Reviewer 1 Report

The manuscript entitled as “Mechanism of action on leukemia cell lines of some derivatives of lupane triterpenes isolated from Phoradendron wattii” by  Lía S. Valencia-Chan et al is very interesting  due to its clinical importance for the clinical practitioners working on the treatment of the leukemia patients. In fact, the current proposal is interesting.  Therefore, I recommend the publication of the present study after some minor revisions. Some suggestions that will fatherly improve the manuscript:

1.       Add one line concise conclusion statement in the abstract section.

2.       Add a few lines on the  leukemia and cite the below manuscript in the introduction section. This will improve the introductory section.

Citation of article. https://doi.org/10.3329/bjms.v18i2.40689

3.       Add future recommendations of the study.

Recheck the spellings and remove the grammatical mistakes

Author Response

  1. Add one-line concise conclusion statement in the abstract section: A concise conclusion line is added in the abstract, as suggested.

  1. Add a few lines on the leukemia and cite the below manuscript in the introduction section. This will improve the introductory section. Citation of article. https://doi.org/10.3329/bjms.v18i2.40689: The recommended article was cited to improve the introductory section.

  1. Add future recommendations of the study: Added future recommendations for the study, in the discussion section

Recheck the spellings and remove the grammatical mistakes.

We recheck the manuscript, thank you very much.

Reviewer 2 Report

This study well complements numerous studies related to the synthesis of promising anticancer drugs based on natural precursors. The work is well planned and organized. However, I have some important comments:

Major

Abstract. Line 19. The authors write about loss of mitochondrial membrane permeability, but their results do not confirm (or disprove) this. You need to rewrite more correctly, based on your own results. I propose to write about the violation of the membrane potential. The same applies to the conclusion section.

In the introduction section, the problem of using native triterpenes as drugs should be described. Indeed, these compounds show low bioavailability due to poor solubility in biological media. This requires modification of their structure.

Line 83. Why were these particular concentrations of the test agent chosen? In addition, different incubation times were used in different experiments (for example, when assessing cytotoxicity and ROS production), this needs to be justified.

The etoposide effect should be mentioned already in the description of the first table. What is the mechanism of its action? This also needs to be mentioned for the convenience of the reader.

Line 115. A decrease in the membrane potential is a signal for the induction of cell death, however, it is nonspecific and is observed not only during apoptosis. Need to rewrite.

Lines 123-126. Why were these targets chosen for analysis? This also needs to be justified.

Figures and tables should be arranged in the order in which they are mentioned. In the presented version, it looks inconvenient.

On fig. 2 panels A-C are not listed. The X-axis and column designations are also not signed. The same applies to Fig. 3

The authors write that they used ANOVA analysis, then I do not quite understand the results described in Fig. 4B. It can be seen that the effects of various agents differ from each other, but the authors do not indicate this. The same applies to Fig. 5.

The discussion looks too mechanical. The following changes are required:

Lines 184-191. In the discussion, the authors should mention the use of mitochondria-targeted molecules to enhance the anti-cancer effect of triterpenes (like TPP, F16, pyridine derivatives, etc.).

Regarding ROS. An important mechanism of their action is lipid peroxidation leading to membrane permeabilization, including mitochondria. This needs to be discussed in the context of the work.

It is known that native triterpenes (betulin, betulinic and betulonic acids) are able to modulate the surface properties of mitochondrial membranes, which was also confirmed on artificial membrane systems (like liposomes). That is, they directly affect the lipid bilayer of membranes. This needs to be discussed in the context of the results obtained by the authors.

In addition, the reason for the increase in ROS levels should be discussed. It is known that triterpenes and their various derivatives (especially those conjugated with mitochondria-targeted delocalized cations) can directly affect the operation of mitochondrial respiratory chain complexes, which leads to ROS overgeneration. This agrees well with the data obtained by the authors.

Please calculate, at least theoretically, the partition coefficient of the obtained compounds (logP). This will assess the ability of agents to interact with membranes.

Minor

Figure 1 looks truncated, please edit.

Line 390 - Should be rhodamine 123.

Author Response

Reviewer 2

  1. Line 19. The authors write about loss of mitochondrial membrane permeability, but their results do not confirm (or disprove) this. You need to rewrite more correctly, based on your own results. I propose to write about the violation of the membrane potential. The same applies to the conclusion section. The recommendation that was given was added
  2. In the introduction section, the problem of using native triterpenes as drugs should be described. Indeed, these compounds show low bioavailability due to poor solubility in biological media. This requires modification of their structure. The recommendation that was given was added.
  3. Line 83. Why were these particular concentrations of the test agent chosen? In addition, different incubation times were used in different experiments (for example, when assessing cytotoxicity and ROS production), this needs to be justified.
  4. Two concentrations were used for compound T1m, IC50 39 μg/mL and 1/2 IC505 μg/mL. Regarding the ROS experiment, it was carried out at an exposure time of less than 18 h, which was the shortest exposure time in the apoptosis experiment, and in the experiment with Rho123, it was carried out at 18 h, as it is an effect consequence of the generation of ROS. This was appended to the manuscript.
  5. The etoposide effect should be mentioned already in the description of the first table. What is the mechanism of its action? This also needs to be mentioned for the convenience of the reader: In the results, the mechanism of action of etoposide is mentioned.
  6. Line 115. A decrease in the membrane potential is a signal for the induction of cell death, however, it is nonspecific and is observed not only during apoptosis. Need to rewrite. In line 126, what is indicated is mentioned; which was rewritten, to improve the manuscript.
  7. Lines 123-126. Why were these targets chosen for analysis? This also needs to be justified. The targets used were justified as recommended
  8. Figures and tables should be arranged in the order in which they are mentioned. In the presented version, it looks inconvenient. The tables and figures were presented in that order because it is the format of the journal.
  9. On fig. 2 panels A-C are not listed. The X-axis and column designations are also not signed. The same applies to Fig. 3: The images were enhanced
  10. The authors write that they used ANOVA analysis, then I do not quite understand the results described in Fig. 4B. It can be seen that the effects of various agents differ from each other, but the authors do not indicate this. The same applies to Fig. 5. In figures 2 to 5, the figure captions were corrected for the significance levels.

The discussion looks too mechanical. The following changes are required:

  1. Lines 184-191. In the discussion, the authors should mention the use of mitochondria-targeted molecules to enhance the anti-cancer effect of triterpenes (like TPP, F16, pyridine derivatives, etc.) The discussion was expanded with respect to the recommendations given regarding the reported activities of triterpenes with respect to their effects on the mitochondrial membrane
  2. Regarding ROS. An important mechanism of their action is lipid peroxidation leading to membrane permeabilization, including mitochondria. This needs to be discussed in the context of the work. Discussion regarding the recommendations given regarding the effects of ROS on lipid peroxidation was expanded, as suggested.
  3. It is known that native triterpenes (betulin, betulinic and betulonic acids) are able to modulate the surface properties of mitochondrial membranes, which was also confirmed on artificial membrane systems (like liposomes). That is, they directly affect the lipid bilayer of membranes. This needs to be discussed in the context of the results obtained by the authors. Discussion about the properties of triterpenes on mitochondria was carried out, as suggested.
  4. In addition, the reason for the increase in ROS levels should be discussed. It is known that triterpenes and their various derivatives (especially those conjugated with mitochondria-targeted delocalized cations) can directly affect the operation of mitochondrial respiratory chain complexes, which leads to ROS overgeneration. This agrees well with the data obtained by the authors. A discussion of adding delocalized cations to mitochondria-targeted triterpenes was made, as suggested.
  5. Please calculate, at least theoretically, the partition coefficient of the obtained compounds (logP). This will assess the ability of agents to interact with membranes. The calculation of the LogP was made and discussed about it, as suggested.

Minor.

  1. Figure 1 looks truncated, please edit: The figure was improved
  2. Line 390 - Should be rhodamine 123: word was changed

Author Response

Review 3:

  1. Title: The title is unclear, rewriting it as a direct and conclusive title would make it easier for the readers to understand.

We changed the title, thank you very much

Lupane triterpene derivatives improve antiproliferative effect on leukemia cells through apoptosis induction.

  1. SHP2, BCL-2, caspase 3, etc should write the full spelling of words in first present.
    The corresponding correction was made

  1. The significance level in the caption should be marked accurately. In figures 2 to 5, the figure captions were corrected for the significance levels.

4.Overall, the experimental design and the results presented in the current manuscript are not sufficient to give a conclusive answer to the research question. To do this, more appropriate experiments shall be done.

We disagree with this comment, we include experiments to clarify antiproliferative effect of a new lupane type triterpene derivatives. In our conclusion we include how a simple derivative can improve the biological activity opening further research. These results are the first structural modification made in tryterpenes in our research group to find a potential molecule for cancer treatment. Future we could include in vitro and in vivo experiments.

Reviewer 4 Report

This study aimed to investigate the antitumor effect and mechanism of action on leukemia cell lines by 23-hydroxy-betulinic acid derivatives. This is continuous research conducted by this team. Based on the present data, the in vitro study suggested that T1m (dimethylated 23-hydroxy betulinic acid) induced antiproliferation by inducing apoptosis, ROS generation, and the loss of mitochondrial membrane permeability. However, the potency of the anti-leukemic activity was weak. Minor structural changes caused by methylation did not significantly improve potency. On the other hand, the target proteins for molecular docking were Bcl-2 and EGFR. However, the Bcl-2 and EGFR were the suspended targets for T1m and T1. Authors are required to present the protein expression of Bcl-2 and Bcl-2/Bax-1 ratio. Or, protein affinity assay of Bcl-2 and EGFR is acceptable to prove the virtual molecular docking prediction. The above viewpoints are the major comments required to address by the authors. The minor comments are listed as follows:

1. please follow the MDPI template.

2. Figures 2 and 3: the context/labels of the X-axis are missing.

3. Figure 1. Chemical structures are not completely shown.

Author Response

Review 4:

This study aimed to investigate the antitumor effect and mechanism of action on leukemia cell lines by 23-hydroxy-betulinic acid derivatives. This is continuous research conducted by this team. Based on the present data, the in vitro study suggested that T1m (dimethylated 23-hydroxy betulinic acid) induced antiproliferation by inducing apoptosis, ROS generation, and the loss of mitochondrial membrane permeability. However, the potency of the anti-leukemic activity was weak. Minor structural changes caused by methylation did not significantly improve potency.

 Reply: The anti-leukemic activity was shown as half maximal inhibitory concentration (IC50) and it is related to potency in inhibiting leukemia cells. In this sense T1m reduced IC50 around 2.5 less compared to original molecule and it also show a better effect in K562 cells compared to non-cancer cells. There are efforts in the scientific community to find not only potency in new molecules but also, security or new targets that can be used alone or in synergy with cancer treatments. We considered our work would contribute with natural products studies in cancer research.

On the other hand, the target proteins for molecular docking were Bcl-2 and EGFR. However, the Bcl-2 and EGFR were the suspended targets for T1m and T1. Authors are required to present the protein expression of Bcl-2 and Bcl-2/Bax-1 ratio. Or, protein affinity assay of Bcl-2 and EGFR is acceptable to prove the virtual molecular docking prediction. The above viewpoints are the major comments required to address by the authors.

 Reply: With all scientific advances there are strategies to follow based on molecular modeling, this was the reason for molecular docking analysis using molecular targets involved in apoptosis for chronic myeloid leukemia to better understand the possible mechanism of action, we show results on Bcl-2 and EGFR. Also molecular docking analysis showed T1m to have higher affinity against BCL-2 and EGFR than its precursor. Molecular docking approach can be used to model the interaction between a molecule and a protein, which allow us to characterize the behavior of molecule in the binding site of target proteins as well as to elucidate fundamental biochemical processes. The second reason was amount of compounds. Unfortunately, we require to obtain enough amount of compound to evaluate protein affinity assay or to evaluate the effect on protein expression, we are working to obtain the compounds. However, this result is a valid contribution.

The minor comments are listed as follows:

  1. please follow the MDPI template. We followed the MDPI template, thank you for your advice.

  1. Figures 2 and 3: the context/labels of the X-axis are missing. The images were enhanced. Thank you for your advice.

  1. Figure 1. Chemical structures are not completely shown. The Figure 1 was enhanced. Thank you for your advice.

Round 2

Reviewer 2 Report

These results can be published, although the mechanism of action of derivatives certainly needs further study.

Minor remark: Ref. [43] does not show a direct effect of triterpenes on lipid membranes. More appropriate Ref. needed (like doi: 10.1016/j.bbamem.2020.183383 or others).

Author Response

These results can be published, although the mechanism of action of derivatives certainly needs further study.

 The title original was: Mechanism of action on leukemia cell lines of some derivatives of lupane triterpenes isolated from Phoradendron wattii has been changed, to be more consistent with what we got in this work: “Lupane triterpene derivatives improve antiproliferative effect on leukemia cells through apoptosis induction.”

Also abstract  and conclusion sections were changed in order to clarify our findings and future research interest.

  • Minor remark: Ref. [43] does not show a direct effect of triterpenes on lipid membranes. More appropriate Ref. needed (like doi: 10.1016/j.bbamem.2020.183383 or others).

The recommended article was cited to improve the discussion section.

Thank you very much for all comments that improved this manuscript.

Reviewer 3 Report

Animal experiments should be added to verify the author's point.

Author Response

Reviewer 3

  • Animal experiments should be added to verify the author's point.

We carefully reviewed manuscript; we change two important parts. We agree with  this reviewer in this point.

We change this asseveration

Abstract

T1m is a potentially promising element in leukemia treatment to this T1m is a potentially promising element for future research

 Conclusion

(T1m) appeared the most promising for possible inclusion in cancer treatments to this (T1m) appeared the most promising for further research

We made those couple of changes, we are very grateful with this comment that help us to improve with facts our conclusion.

By the other hand as we mentioned before, natural product studies have limitations one of them are low amount of compounds isolated from natural resources and the difficulty to synthesize them, these limitations are known for scientific community. In this work we have very interesting results, that show how very small modifications can improve effects on leukemia cells, in this last manuscript version we include is necessary further research. However, our results are value to be published in order to share our advances to scientific community.

And again we are working to isolated more compounds. In the future we could include in vitro and in vivo experiments, to propose a potential leukemia treatment. Thank you very much for your value review.
